# The Burden on Cohabitants of Patients with Chronic Spontaneous Urticaria: A Cross-Sectional Study

**DOI:** 10.3390/jcm11113228

**Published:** 2022-06-06

**Authors:** Manuel Sánchez-Díaz, Maria-Carmen Salazar-Nievas, Alejandro Molina-Leyva, Salvador Arias-Santiago

**Affiliations:** 1Dermatology Unit, Hospital Universitario Virgen de las Nieves, IBS Granada, 18002 Granada, Spain; manolo.94.sanchez@gmail.com (M.S.-D.); msalazarn@hotmail.com (M.-C.S.-N.); salvadorarias@ugr.es (S.A.-S.); 2Urticaria Clinic, Hospital Universitario Virgen de las Nieves, 18002 Granada, Spain; 3Dermatology Department, School of Medicine, University of Granada, 18002 Granada, Spain

**Keywords:** urticaria, cohabitants, quality of life, anxiety, type D personality

## Abstract

Chronic Spontaneous Urticaria (CSU) has been associated with patients’ poor quality of life. Despite being a chronic disease that could alter the quality of life of the people who live with patients, the potential burden on their cohabitants has not been studied to date. The aim of this study is to analyze the relationship between the patient’s quality of life, disease control, disease duration and family quality of life and the cohabitant’s mood disturbances, sexual dysfunction, type D personality and sleep quality. A cross-sectional study including patients suffering from CSU and their cohabitants was performed. Sociodemographic variables and disease activity, quality of life, sleep, sexual disfunction, anxiety, depression and type D personality were collected using validated questionnaires. Sixty-two subjects, 31 CSU patients and 31 cohabitants, were included in the study. Worse disease control and poorer quality of life in patients were associated with poorer family quality of life and higher rates of anxiety among the cohabitants (*p* < 0.05). Patients’ poor quality of life was associated with reduced sexual satisfaction among the cohabitants (*p* < 0.05). Long disease duration (>10 years) was associated with an increased prevalence of type D personality among the cohabitants (prevalence ratio: 2.59, CI 95% 1.03–7.21). CSU seems to have an impact on the quality of life of cohabitants, especially in terms of increased rates of anxiety, poorer quality of life and reduced sexual satisfaction. The prolonged course of the disease could be associated with the increased presence of non-adaptative personality traits.

## 1. Introduction

Chronic spontaneous urticaria (CSU) is characterized by urticarial lesions for a period of time exceeding six weeks [1]. This disease generates symptoms most days and does not have any detectable trigger, as most cases are thought to be idiopathic [2]. Although it is usually self-limiting, generally lasting 2 to 5 years, a prolonged course for a longer time is possible. Since CSU skin lesions are visible and often accompanied by intense pruritus, CSU has a profound impact on the patient’s quality of life [3,4,5], particularly in terms of higher rates of mood status disturbances, difficulties in social relationships and poorer personal self-esteem [6].

This impairment on quality of life has also been described for other skin diseases, such as acne [7,8], psoriasis [9] and hidradenitis suppurativa [10]. Most of the available scientific evidence holds that quality of life deficits in skin diseases are driven by the symptomatology of the different diseases, the duration of the disease and individual factors such as personality traits [11,12]. Despite the impact of CSU on patients’ quality of life having been addressed in previous studies [13,14], there is still a lack of evidence on the impact that CSU can have on those living with patients. Similar to what has already been shown to occur in other skin diseases such as acne [15], psoriasis [16] and hidradenitis suppurativa [17], the patient’s symptomatology, decreased quality of life, higher rates of anxiety and depression and sexual dysfunction could result in a negative impact on cohabitants. This could include a poorer quality of life for cohabitants, mood disorders, sexual dysfunction, sleep disturbances or personality disorders.

Identifying the factors associated with a poorer quality of life of those living with the patient, as well as those factors related to mood status disturbances or dysfunctional personality traits such as type D personality (TDp) [18], would be useful in order to develop a holistic approach to the disease. This approach should take into account not only the patient’s symptomatology and quality of life but also the impact these factors have on their family background.

### Objectives of the Study

The aims of this study are: (a) to evaluate the quality of life, mood disturbances and sexual dysfunction among cohabitants of patients with CSU; (b) to analyze the relationship between the patient’s quality of life and family quality of life and the cohabitant’s mood disturbances, sexual dysfunction, TDp and sleep quality; (c) to evaluate the association between the patient’s disease control, disease duration and family quality of life and the cohabitant’s mood disturbances, sexual dysfunction, TDp and sleep quality.

## 2. Materials and Methods

### 2.1. Design

A cross-sectional study including patients suffering from mild to severe CSU cases and their cohabitants was conducted in order to evaluate the potential impact of CSU on different aspects of the quality of life of cohabitants.

### 2.2. Patients

The subjects included in the study were recruited from two sources: (a) patients who received health care in the Urticaria Clinic of the Virgen de las Nieves University Hospital and their cohabitants. Both patients and cohabitants were offered to complete an online questionnaire after their protocolized follow-up consultation. (b) Patients who were contacted by e-mail by “Asociación de Afectados por Urticaria Crónica”, the official Spanish patient association for patients with CSU, and their cohabitants. These subjects were offered to complete the online version of the questionnaire. The patients were recruited between September 2021 and March 2022. The questionnaires for the two sources of subjects were identical and were completed online.

### 2.3. Inclusion Criteria

The inclusion criteria for the patients were as follows: (a) patients with a clinical diagnosis of CSU of all degrees of severity and with any type of treatment; (b) patients aged 18 years old or older; (c) informed consent from the patient to be included in the study. The inclusion criteria for the cohabitants were as follows: (a) cohabitants of patients with CSU who had agreed to participate in the study; (b) the cohabitant had to be the patient’s partner, regardless of their marital status; (c) cohabitants aged 18 years old or older; (d) informed consent from the cohabitant to be included in the study.

### 2.4. Exclusion Criteria

The exclusion criteria were: (a) refusal from the patient or cohabitant to participate in the study; (b) patients or cohabitants who suffered from any other major disease that could affect their quality of life. The considered diseases included: (i) active oncological diseases; (ii) any cardiopulmonary, neurologic, digestive, metabolic, musculoskeletal or urinary disease limiting daily activity or generating significant symptomatology; (iii) psychiatric disorders existing prior to the onset of urticaria; (iv) skin diseases other than urticaria that cause significant impairment of one’s quality of life.

### 2.5. Ethics

The present study was approved by the Research Ethics Committee of the “Hospital Universitario Virgen de las Nieves” (internal code: 2367-N-21) and is in accordance with the Declaration of Helsinki.

### 2.6. Variables of Interest

#### 2.6.1. Main Variables

The main variables included variables related to the severity of the disease and those related to the assessment of the quality of life:(1)Variables related to the severity and characteristics of the disease in patients:Urticaria Control Test: It consists of eight different questions about the physical and quality-of-life symptoms related to urticaria in the 4 weeks prior to the consultation. The questions are answered using a Likert scale that values the severity of the urticaria from 0 to 4. The values obtained range from 0 (indicating no control) to 32 (indicating total control) [19].The age of onset, evolution time of the disease, date of diagnosis and current treatments were collected.(2)Variables related to quality of life, anxiety and depression, sleep disturbance, sexual dysfunction and TDp in both patients and cohabitants. The following validated questionnaires were collected:Dermatology Life Quality Index (DLQI): It is an indicator of the general dermatologic quality of life in patients over 16 years of age. The questionnaire consists of 10 questions that are each scored on a Likert scale from 0 to 3, with a final score of 0 being the least affected and a final score of 30 being the most affected. The questions refer to the last 7 days [20].Chronic Urticaria Quality of Life Questionnaire (CUQoL): It is a questionnaire that includes physical, emotional and social characteristics and aspects of the urticaria itself. It consists of 23 Likert-type questions that are evaluated from 1 (never) to 5 (very much), finally obtaining a range of 0 (no quality-of-life impairment) to 100 (maximum quality-of-life impairment). Different subscales are collected in this questionnaire, including pruritus, swelling, impact on daily activities, sleep disturbances, daily limitations and physical aspects, as well as the overall CUQoL score [9].Family Dermatology Life Quality Index (FDLQI): It is an indicator of the general quality of life among cohabitants of patients suffering from skin diseases. The questionnaire consists of 10 questions that are scored on a Likert scale from 0 to 3 each, with 0 being the least affected and 30 being the most affected. The questions refer to the last 7 days [21].Hospital Anxiety and Depression Scale (HADS): This validated questionnaire is composed of 14 statements in which the patient must show the degree of agreement/disagreement, scoring each question using an adapted Likert scale. It is subdivided into two scales, with odd-numbered questions being scored for anxiety and even-numbered questions being scored for depression. A score ≥ 8 on any of the subscales was considered indicative of anxiety or depression, respectively [22].DS14 Questionnaire: It was used to evaluate the presence of TDp. TDp is described as the presence in the same individual of traits of negative affectivity and social inhibition. It consists of a Likert-type questionnaire composed of 14 items, 7 for negative affectivity and 7 for social inhibition. Each response is answered with values between 0 (completely false) and 4 (completely true). A score ≥ 10 in both spheres is established as a cut-off point as an indicator of TDp [23,24].International Index of Erectile Function (IIEF-5) [25] and Female Sexual Function Index (FSFI-6) [26] questionnaires: They were used to collect data on sexual dysfunction in men and women. The IIEF-5 covers all five spheres of sexual function in the males, and a score ≤ 21 was considered significant. The FSFI-6 assesses the six items of female sexual function, and a score ≤ 19 was established as indicative of dysfunction.Pittsburgh Sleep Quality Index (PSQI) Questionnaire: This is a validated questionnaire to study the patients’ quality of sleep. It consists of different questions in which the patient must mark one of the multiple answers offered. The global score is scored from 0–21 points, with 21 being the number that implies the greatest impairment of sleep quality. A global score greater than five is considered relevant from the point of view of sleep quality impairment [27].Numeric Rating Scale (NRS) for sexual impairment: The patients and cohabitants had to choose, from a scale of 1–10, the degree to which their sexual impairment was associated with their CSU, as has been previously reported [28].

#### 2.6.2. Other Variables

Sociodemographic, biometric and clinical variables including age, sex, comorbidities, previous treatments for CSU, marital status and educational level were recorded by questionnaires. Specific questions regarding all the exclusion criteria were included, which were mandatory to answer.

### 2.7. Statistical Analysis

Descriptive statistics were used to evaluate the characteristics of the sample. The Shapiro–Wilk test was used to assess the normality of the variables. The continuous variables are expressed as the mean and standard deviation (SD). The qualitative variables are expressed as the relative and absolute frequency distributions. The χ^2^ test or Fisher’s exact test, as appropriate, were used to compare the nominal variables, and Student’s *t*-test or the Wilcoxon–Mann–Whitney test were used to compare between the nominal and continuous data. To explore the possible associated factors, a simple linear regression was used for the continuous variables. *p* values of less than 0.05 were considered statistically significant. The statistical analyses were performed using JMP version 14.1.0 (SAS institute, Cary, NC, USA).

## 3. Results

### 3.1. Sociodemographic and Clinical Features of the Sample

Sixty-two subjects were included in the study: thirty-one CSU patients and their corresponding thirty-one cohabitants. The participants were contacted during clinical care at the Urticaria Clinic (54.8%, 17/31) or through the patient association (45.2%, 14/31). There were no differences in terms of age, sex, disease control, disease duration or occupation between the patients and cohabitants recruited by both sources (*p* > 0.60).

#### 3.1.1. Patients’ Characteristics

The mean age of the patients was 46.41 (SD 8.92), and the female-to-male ratio was 1.81. Most of the patients had had an active job (74.2%, 23/31) and professional or university studies (80.6%, 25/31). The mean evolution time of the disease in the sample indicated was 10.70 (SD 11.67), with 38.7% (12/31) of the patients having a long-lasting disease (>10 years).

The mean values and outcomes of the quality-of-life questionnaires can be seen in Table 1. The prevalence of anxiety and depression in the patients was 38.7% (12/31) and 48.4% (15/31), respectively. Sexual dysfunction was detected in 58% of the patients (18/31): 60% (12/20) of the females and 54.5% (6/11) of the males.

#### 3.1.2. Cohabitants’ Characteristics

The mean age of the cohabitants was 45.67 (SD 10.87), and the female-to-male ratio was 0.93. Similar to what was found for the patients, most of the cohabitants had an active job (64.5%, 20/31) and professional or university studies (61.3%, 19/31).

The mean values and outcomes of the quality-of-life questionnaires can be seen in Table 2. The prevalence of anxiety and depression in the cohabitants was 22.6% (7/31) and 16.1% (15/31), respectively, which were lower than those found in the patients (*p* = 0.04). Sexual dysfunction was detected in 41.9% of the cohabitants (13/31): 60% (9/15) of the females and 25% (4/16) of the males. The cohabitants’ family-quality-of-life scores (FDLQI), anxiety and depression were found to be not associated with the cohabitants’ age, gender, educational level or occupation (*p* > 0.30).

### 3.2. Association of Patients’ Quality-of-Life Indexes with Quality-of-Life Indexes, Mood Disturbances and Sexual Function in Cohabitants

The association between the quality-of-life indexes of the patients (DLQI and CUQOL) and the quality-of-life indexes, mood disturbances, sexual disfunction and TDp of the cohabitants was explored (Table 3). It was observed that patients’ poorer quality of life—measured by both indexes, DLQI and CUQOL—was associated with higher rates of anxiety in the cohabitants (*p* = 0.04 for DLQI; *p* = 0.06 for CUQOL), a poorer quality of life of the family (*p* = 0.005) and lower sexual satisfaction among the cohabitants (NRS for sexual impairment, *p* < 0.05). In contrast, no significant associations were found for the cohabitants’ depression, TDp, sleep quality or specific sexual dysfunction indexes.

### 3.3. Association of Patients’ Disease Control with Quality-of-Life Indexes, Mood Disturbances and Sexual Function in Cohabitants

The association between worse disease control (UCT) and the quality-of-life indexes, mood disturbances, sexual disfunction and TDp of the cohabitants was explored (Table 4). It was found that worse disease control was associated with poorer family quality of life (*p* < 0.0001) and higher rates of cohabitants’ anxiety (*p* = 0.045). No significant associations were found for cohabitants’ depression, TDp, sexual dysfunction or sleep disturbances.

### 3.4. Association of Patients’ Disease Duration with Quality-of-Life Indexes, Mood Disturbances and Sexual Function in Cohabitants

The association between disease duration (years) and the quality-of-life indexes, mood disturbances, sexual disfunction and TDp of the cohabitants was explored (Table 4). It was found that a longer patient disease duration correlated with higher anxiety rates (*p* = 0.05) and higher rates of TDp (*p* = 0.04) among the cohabitants. After a contingency analysis was performed, it was confirmed that a long duration of CSU (>10 years vs. <10 years) was associated with higher rates of TDp in the cohabitants. Moreover, for this group, a 2.59-times increase in the prevalence of TDp was found (prevalence ratio: 2.59, CI 95% 1.03–7.21). In this case, no associations were found for family quality of life, depression or sexual impairment.

## 4. Discussion

CSU is a chronic skin disease which has been previously associated with the poor quality of life of patients [6,29,30]. However, there are currently no studies assessing the impact that this disease may have on the patient’s cohabitants. This study explored, for the first time, the burden of the disease on those living with patients affected by CSU, showing that worse disease control and poorer quality of life in patients seem to be associated with poorer family quality of life and higher rates of anxiety in cohabitants. In addition, the prolonged course of the disease seems to be associated with higher rates of tTDp in cohabitants, as well as higher rates of anxiety.

There are several factors that can contribute to the impact of a disease on those living with the patients. First of all, worse disease control has been associated with poorer quality of life among cohabitants in a variety of medical conditions, such as inflammatory bowel disease [31], psoriasis [16] and hidradenitis suppurativa [17]. On the other hand, the diseases themselves can cause alterations in the mood and personality traits of the patients [5,18,32], which could favor the negative impact on the social network close to the patient.

In this context, the results of the present study show that, despite the fact that anxiety is more frequent among patients than among their cohabitants, patients’ poor quality of life, long disease duration and worse disease control are significantly associated with increasing rates of anxiety among cohabitants. No previous studies have demonstrated this association between patients’ quality of life and disease control and the mood status alterations of cohabitants in dermatologic disorders.

Our study showed that the cohabitants of patients who have suffered from CSU for longer periods of time are more likely to meet the criteria for TDp. TDp can be defined as the combination of social inhibition and negative affectivity. Social inhibition can be defined as the tendency to withdrawal from new people and to avoid social situations, whereas negative affectivity is defined as the tendency to experience negative emotions [24]. Both characteristics are related to dysfunctional coping strategies. As a chronic disease, CSU could play an important role in the development of dysfunctional and negative ways of coping with reality among those living with patients with CSU. Since TDp has been associated with cardiovascular disorders [33,34] and poorer outcomes in a variety of diseases [32,35,36], longer durations of urticaria (>10 years) could pose a risk to the physical and mental well-being of cohabitants.

The relationship between quality of life and personality has been widely studied for other diseases, such as acne [7], psoriasis [37] and skin cancer [38]. Whether the personality traits found in these studies are the cause of the low quality of life or the consequence of the disease condition is a matter to be clarified in future studies. Prospective studies addressing the development of specific personality traits such as TDp over time among patients suffering from skin diseases—as well as among their cohabitants—would be of interest to elucidate this question.

Finally, the relationship between the quality of life of patients and the sexual dysfunction of cohabitants has been explored in the present study. Although no differences were found in terms of validated scales such as FSFI or IIEF, we found correlations between the HRQOL measures (DLQI and CUQOL) and NRS for sexual impairment among the cohabitants. Poor physical health has been consistently associated with sexual dysfunction in the general population [39], whereas the impact of age might be unclear [40]. Studies that include a larger sample size and stratification by age would be of interest to unequivocally identify the relationship between patients’ quality of life and the sexual dysfunction of cohabitants, avoiding confounding biases.

## 5. Limitations

The main limitations of the present study are: (a) the sample size, which could have limited the detection of significant differences; (b) the cross-sectional design, which makes it impossible to assess causality; and (c) the inclusion of only the patients’ partners, without including other types of cohabitants.

## 6. Conclusions

CSU seems to have an impact on the quality of life among cohabitants, especially in terms of increased rates of anxiety, poorer quality of life and reduced sexual satisfaction. In addition, the prolonged course of the disease could be associated with substantial changes in the way cohabitants cope with life, increasing the presence of non-adaptative personality traits such as TDp. A holistic approach to this pathology—taking into account not only the patient’s manifestations but also those of the people living with him/her—will improve the medical care of CSU.

## Figures and Tables

**Table 1 jcm-11-03228-t001:** Sociodemographic features of the patients, characteristics of the disease and quality-of-life indicators.

Variables Patients (*n* = 31)					
Sociodemographic features					
Age (years)	46.41 (SD 8.92)		Occupation	Employed	74.2% (23/31)
				Unemployed	25.8% (8/31)
Sex (%)	Male:	35.5% (11/31)	Educational level	No studies or compulsory education	19.4% (6/31)
	Female:	64.5% (20/31)		Professional or university studies	80.6% (25/31)
**Disease characteristics**					
Disease duration (years)	10.70 (SD 11.67)		Urticaria Control Test score	16.29 (SD 6.73)	
Disease duration	<10 years	61.3% (19/31)	Current treatment for CSU	Antihistamines	61.3% (19/31)
	>10 years	38.7% (12/31)		Omalizumab	38.7% (12/31)
Quality-of-life indicators					
DLQI	10.35 (SD 7.24)		Overall CUQOL	33.45 (SD 21.61)	
DS14 (% of positive test)	29% (9/31)		PSQI	10.16 (SD 4.53)	
HADS depression (% of positive test)	48.4% (15/31)		HADS Anxiety (% of positive test)	38.7% (12/31)	
FSFI (% of female sexual dysfunction)	60% (12/20)		IIEF (% of male sexual dysfunction)	54.5% (6/11)	

CUQOL: Chronic Urticaria Quality of Life questionnaire; DLQI: Dermatology Quality of Life Index; DS14: Questionnaire for Type D personality; FSFI: Female Sexual Funcion Index; HADS: Hospital Anxiety and Depression Scale; IIEF: International Index of Erectile Funcion; PSQI: Pittsburg Sleep Quality Index; SD: Standard desviation.

**Table 2 jcm-11-03228-t002:** Sociodemographic features of the patients, characteristics of the disease and quality-of-life indicators.

Variables Cohabitants (*n* = 31)					
Sociodemographic features					
Age (years)	45.67 (SD 10.87)		Occupation	Employed	64.5% (20/31)
				Unemployed	35.5% (11/31)
Sex (%)	Male:	51.6% (16/31)	Educational level	No studies or compulsory education	38.7% (12/31)
	Female:	48.4% (15/31)		Professional or university studies	61.3% (19/31)
**Quality-of-life indicators**					
FDLQI	8.35 (SD 6.15)		PSQI	5.32 (SD 3.04)	
DS14 (% of positive test)	35.48% (11/31)		HADS Anxiety (% of positive test)	22.6% (7/31)	
HADS depression (% of positive test)	16.1% (15/31)		IIEF (% of male sexual dysfunction)	25% (4/16)	
FSFI (% of female sexual dysfunction)	60% (9/15)				

FDLQI: Family Dermatology Quality of Life Index; DS14: Questionnaire for Type D personality; FSFI: Female Sexual Funcion Index; HADS: Hospital Anxiety and Depression Scale; IIEF: International Index of Erectile Funcion; PSQI: Pittsburg Sleep Quality Index; SD: Standard desviation.

**Table 3 jcm-11-03228-t003:** Correlation between the quality-of-life indexes of the cohabitants and the patients’ DLQI and CUQOL.

Factors	Patient DLQI		Patient CUQOL	
	Mean/Beta	*p* Value	Mean/Beta	*p* Value
FDLQI	0.41 (SD 0.13)	0.005	0.14 (SD 0.04)	0.004
Cohabitant anxiety (HADS-A ≥ 8)	Yes: 14.42 (SD 2.64)	0.04	Yes: 44.56 (SE 7.97)	0.06
	No: 9.16 (SD 1.43)		No: 30.21 (SE 4.30)	
Cohabitant depression (HADS-D ≥ 8)	Yes: 13.80 (SD 3.22)	0.25	Yes: 40 (SE 9.74)	0.47
	No: 9.69 (SD 1.41)		No: 32.19 (SE 4.27)	
Cohabitant type D personality	Yes: 10.63 (SD 2.22)	0.87	Yes: 30.92 (SD 6.6)	0.63
	No: 10.20 (SD 1.64)		No: 34.83 (SD 4.89)	
Cohabitant NRS for sexual impairment	0.17 (SD 0.008)	0.043	0.06 (SD 0.02)	0.047
Male cohabitant sexual impairment index (Men–IIEF)	0.01 (SD 0.14)	0.90	0.03 (SD 0.04)	0.47
Female cohabitant sexual impairment index (Women–FSFI)	0.02 (SD 0.41)	0.94	0.02 (SD 0.14)	0.82
Cohabitant PSQI	0.007 (SD 0.7)	0.92	0.03 (SD 0.025)	0.33

CUQOL: Chronic Urticaria Quality of Life questionnaire; FDLQI: Family Dermatology Quality of Life Index; FSFI: Female Sexual Funcion Index; HADS: Hospital Anxiety and Depression Scale (A—Anxiety; D—Depression); IIEF: International Index of Erectile Funcion; NRS: Numeric Rating Scale; PSQI: Pittsburg Sleep Quality Index; SD: Standard desviation.

**Table 4 jcm-11-03228-t004:** Association of patients’ disease control and disease duration with quality-of-life indexes, mood disturbances, TDp and sexual disfunction in cohabitants.

Factors	Disease Control (Urticaria Control Test)		Disease Duration (Years)	
	Mean/Beta	*p* Value	Mean/Beta	*p* Value
FDLQI	−0.61 (SD 0.12)	<0.0001	0.10 (SD 0.09)	0.27
Cohabitant anxiety (HADS-A ≥ 8)	Yes: 18.05 (SD 1.36)	0.045	Yes: 15.85 (SE 4.35)	0.05
	No: 11.71 (SD 2.63)		No: 9.20 (SE 2.35)	
Cohabitant depression (HADS-D ≥ 8)	Yes: 15.40 (SD 3.05)	0.75	Yes: 11.40 (SE 5.30)	0.88
	No: 16.46 (SD 1.34)		No: 10.57 (SE 2.32)	
Cohabitant type D personality	Yes: 15.63 (SD 2.05)	0.87	Yes: 16.45 (SD 3.32)	0.04
	No: 16.65 (SD 1.52)		No: 7.55 (SD 42.46)	
Cohabitant NRS for sexual impairment	−0.19 (SD 0.15)	0.69	0.009 (SD 0.05)	0.87
Male cohabitant sexual impairment index (Men–IIEF)	−0.03 (SD 0.14)	0.90	0.04 (SD 0.11)	0.67
Female cohabitant sexual impairment index (Women–FSFI)	−0.64 (SD 0.40)	0.13	0.04 (SD 0.20)	0.84
Cohabitant PSQI	−0.05 (SD 0.08)	0.52	0.08 (SD 0.04)	0.08

FDLQI: Family Dermatology Quality of Life Index; FSFI: Female Sexual Funcion Index; HADS: Hospital Anxiety and Depression Scale (A—Anxiety; D—Depression); IIEF: International Index of Erectile Funcion; NRS: Numeric Rating Scale; PSQI: Pittsburg Sleep Quality Index; SD: Standard desviation.

## Data Availability

The data presented in this study are available on reasonable request from the corresponding author.

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
