# Peer review of "The Burden on Cohabitants of Patients with Chronic Spontaneous Urticaria: A Cross-Sectional Study"

_jcm, 2022, doi:10.3390/jcm11113228_

Round 1
Reviewer 1 Report
- Introduction should be expanded . This section of the manuscript should be expanded with more scientific information and references.
- Please mention in the introduction about other dermatological disease, in which there is also the problem of the low quality of life. Please use:
Chilicka, K., Rogowska, A. M., Szyguła, R., & Taradaj, J. (2020). Examining Quality of Life After Treatment with Azelaic and Pyruvic Acid Peels in Women with Acne Vulgaris. Clinical, cosmetic and investigational dermatology, 13, 469–477. https://doi.org/10.2147/CCID.S262691
Chilicka, K., Rogowska, A. M., Szyguła, R., & Adamczyk, E. (2020). Association between Satisfaction with Life and Personality Types A and D in Young Women with Acne Vulgaris. International journal of environmental research and public health, 17(22), 8524. https://doi.org/10.3390/ijerph17228524
2. Line 55-58 shouldn't be in section introduction. It should be a separate chapter or subsection.
3. The discussion should be also expanded- more researches and references
4. The study limitation should be separate subsection.
5. 29 references is too less for article.
Congratulations on taking up the topic, the quality of life of patients is an extremely important aspect of their life. I am looking forward to see the research with more respondents in the future.
Author Response
Dear Reviewer,
Thank you for your comments, as they allow us to improve the scientific quality of our research.
1) Introduction has been expanded to include some new information.
2) Other skin diseases with low quality of life have been mentioned, including acne, psoriasis and hidradenitis suppurativa.
2) The objectives of the study have been included in a separate subsection.
3) Discussion has been expanded.
4) Limitations has been included in a separate section.
5) The number of references has been increased.
Reviewer 2 Report
- Manuscript should be written coherently, e.g. The word ‘TDp’ (line no. 107) should be abbreviated where first used, the definition of same to be provided in section 2.6 (Variable of Interest). In manuscript TDp is defined in the discussion section.
- There are many typological errors throughout manuscript for e.g. line number 110-111; “The questionnaire consists of 10 questions that are scored on a Likert scale from 0 to 3 each, with 0 being the least affected and 30 the most affected.” It is misleading, 3 or 30??
- Correlation between quality-of-life indexes of the cohabitants and patient’s DLQI and 215 CUQOL for. e.g. FSFI (% of female sexual dysfunction) and IIEF (% of male sexual dysfunction) reported no significant difference. Whereas Cohabitant NRS for sexual impairment was found to be significant. Since the mean age for Cohabitants was 45.67 years and for patient was 46.41 years. There are studies where sexual impairment was reported even in healthy subjects with age of 40 or above. How authors can address this bias in context with their study. Please cite appropriate reference and incorporate same in the discussion section.
- Line no. 239-241 “Authors should discuss the results and how they can be interpreted from the perspective of previous studies and of the working hypotheses. The findings and their implications should be discussed in the broadest context possible. Future research directions may also be highlighted” such instruction to write discussion is mandatory as part of journal guildelines?
- There are some limitations like inadequate sample size and cohabitants selection criteria, which limited the conclusion of the study.
Author Response
Dear Reviewer,
Thank you very much for your comments, as they allow us to improve the scientific quality of our research. Below there is a point-by-point response to your suggestions:
1) The mistake regarding type D personality abbreviation has been corrected. As well, the definition has been included in the suggested section.
2) It is not an error, 0 is the global better score, and 30 the worse global score (there are 10 questions). It has been rephrased to make it simple to understand.
3) As you have suggested, there is a possible confusion bias caused by age. However, as it has been included in the discussion section, the relationship between age and sexual dysfunction is not yet fully elucidated as a clear independent associated factor (i.e. the association between the two could also be mediated by confounding factors such as increased illness at older ages). However, this has been included in the discussion section, and age- and sex-stratified analyses are proposed for future studies.
4) This is indeed an error, this part of the statement was not removed when the survey data was transferred to the template provided by the journal.
5) The suggested limitations are included in the "limitations section".
Reviewer 3 Report
The interesting topic of the paper falls within the scope of JCM.
The proposed hypotheses are important with high scientific soundness.
The experimental and analysis methodology is suitable with high feasibility.
There are sufficient details given to replicate the proposed analysis.
Some inappropriate self-citations by authors are detected. The 8th, 9th and 10th references are not justified by the sentence: „Similar to what has already been shown to occur in other skin diseases such as acne (8), psoriasis (9), or hidradenitis suppurativa (10)”. The 11th, 21st and 25th citations are not justified also.
Please change the text: „The β coefficient and SD were used to predict the log odds of the dependent variable. Significantly associated variables ( p < 0.05) or those showing trends towards the statistical significance ( p < 0.20) were included in the multivariate analysis.” It is similar to:
https://www.ncbi.nlm.nih.gov/pmc/articles/PMC8949115//
Author Response
Dear Reviewer,
Thank you for your comments, as they allow us to improve the scientific quality of our research. Below, there is a point-by-point response to your suggestions.
1) Some of the references (those related to studies of quality of life in patients) have been modified. However, the references regarding quality of life in cohabitants are adequate, as there are currently few studies that assess quality of life in cohabitants, several of them from the research group associated with our hospital.
2) The text has been changed to be different to the cited article.